# Healthy Food Retail during the COVID-19 Pandemic: Challenges and Future Directions

**DOI:** 10.3390/ijerph17207397

**Published:** 2020-10-11

**Authors:** Lucia A. Leone, Sheila Fleischhacker, Betsy Anderson-Steeves, Kaitlyn Harper, Megan Winkler, Elizabeth Racine, Barbara Baquero, Joel Gittelsohn

**Affiliations:** 1School of Public Health and Health Professions, University at Buffalo, Buffalo, NY 14213, USA; 2Law Center, Georgetown University, Washington, DC 20001, USA; sef80@georgetown.edu; 3Department of Nutrition, University of Tennessee, Knoxville, TN 37996, USA; Eander24@utk.edu; 4Bloomberg School of Public Health, Johns Hopkins University, Baltimore, MD 21205, USA; kharpe14@jhu.edu (K.H.); jgittel1@jhu.edu (J.G.); 5School of Public Health, University of Minnesota, Minneapolis, MN 55455, USA; mwinkler@umn.edu; 6Department of Public Health Sciences, College of Health and Human Services University of North Carolina at Charlotte, Charlotte, NC 28223, USA; efracine@uncc.edu; 7Department of Health Services School of Public Health, University of Washington Seattle, WA 98195, USA; bbaquero@uw.edu

**Keywords:** retail food environment, food purchasing, federal nutrition assistance, COVID-19, grocery stores, restaurants, dietary intake

## Abstract

Disparities in dietary behaviors have been directly linked to the food environment, including access to retail food outlets. The Coronavirus Disease of 2019 (COVID-19) pandemic has led to major changes in the distribution, sale, purchase, preparation, and consumption of food in the United States (US). This paper reflects on those changes and provides recommendations for research to understand the impact of the pandemic on the retail food environment (RFE) and consumer behavior. Using the Retail Food Environment and Customer Interaction Model, we describe the impact of COVID-19 in four key areas: (1) community, state, tribal, and federal policy; (2) retail actors, business models, and sources; (3) customer experiences; and (4) dietary intake. We discuss how previously existing vulnerabilities and inequalities based on race, ethnicity, class, and geographic location were worsened by the pandemic. We recommend approaches for building a more just and equitable RFE, including understanding the impacts of changing shopping behaviors and adaptations to federal nutrition assistance as well as how small food business can be made more sustainable. By better understanding the RFE adaptations that have characterized the COVID-19 pandemic, we hope to gain greater insight into how our food system can become more resilient in the future.

## 1. Introduction

In the United States (US), substantial socioeconomic and racial disparities exist in dietary behaviors [1]. Limited access to fresh food, coupled with a greater prevalence of fast food outlets in lower-income and minority neighborhoods, is partially responsible for sub-optimal eating patterns among residents [2]. The Coronavirus Disease of 2019 (COVID-19) pandemic has placed unprecedented strain on the US food system and changed the way food is distributed, sold, obtained, prepared, and consumed [3,4]. In the early weeks of the pandemic, grocery retailers saw overwhelming demand paired with panic buying resulting in empty shelves [5]. Restaurants have been temporarily closed in many communities and some have even permanently closed as they were unable to weather the financial burden of the temporary closures and/or the required additional pandemic safeguards [6]. To reduce risk of exposure, many consumers shifted to online food shopping and opted for curbside pickup or home delivery over entering retail stores [7]. Changes in consumer purchasing, coupled with government-mandated business closures, also negatively impacted food growers and producers. Due to the lack of demand from restaurants, there were reports of farmers who found it more economical to plow under crops and cull their herds [8,9,10,11,12].

The virus has disproportionality impacted food access for groups that already had higher rates of food insecurity (see Table 1) [13]. COVID-19 has also further exacerbated existing disparities as coping strategies (e.g., bulk purchasing, online ordering, food delivery) are largely unavailable to those with already limited food access [11,14]. Food insecurity disproportionately affects communities who have been historically oppressed, most notably communities of color, due to policies and structures obstructing access to affordable foods [15]. Individuals in these communities often do not have equal access to resources and are more likely to have lost jobs during the COVID-19 crisis, leading to a further increased risk of food insecurity [16]. Before the pandemic, 21% of Non-Hispanic Black households experienced food insecurity [17]; currently, that proportion is estimated at 38% and will likely continue to rise the longer the pandemic persists and during the resulting economic recovery [18]. Many communities have also been impacted by uprisings against police brutality and structural racism that may have damaged, disrupted, or destroyed food retail outlets and other infrastructure, creating even more food access issues [19].

Despite the many possible effects of the pandemic on components of the retail food environment (RFE), no literature exists that explores these effects in a systematic manner, and then uses these findings to suggest and prioritize next steps. To address this gap, we used the Retail Food Environment and Customer Interaction Model developed by Winkler and colleagues (also in this issue) to describe the impact of COVID-19 on the US RFE in four key areas outlined by the model: (1) community, state, tribal, and federal policy; (2) retail actors, business models, and retail sources; (3) customer experiences in retail setting; and (4) customer dietary intakes. This new model is the first attempt to describe the role of RFE on diet, and the unprecedented change in RFE due to the pandemic allowed an opportunity to both test the new model and to systematically structure our paper. For the purposes of the model and this paper, the RFE includes food stores (grocery, supermarket), food service (restaurants, institutional food), and emergency food (food pantries, food banks). In an effort to build a stronger, more sustainable food system for the future, we also identified research priorities and strategic programmatic directions related to the RFE in the pandemic context.

## 2. Community, State, Tribal, and Federal Policy Affecting the Retail Food Environment

During the pandemic, a variety of macrolevel factors at the community, state, tribal, federal, and global levels have influenced the RFE and customers’ behaviors. Table 2 lists a range of COVID-19-relevant US federal government responses, including policies, programs, and operational guidelines related to food distribution and donations, household food handling and eating out, federal nutrition assistance, and federal nutrition education and promotion. The key US government departments and agencies charged with RFE-related pandemic responses include the Department of Agriculture (USDA), the Department of Commerce, the Department of Health and Human Services, Centers for Disease Control and Prevention (CDC), the Food and Drug Administration (FDA), and the Department of Homeland Security, specifically the Federal Emergency Management Agency (FEMA). Policies targeting retailers included new guidance from the CDC related to safe operations, a Paycheck Protection Program (PPP) which provided forgivable business loans, and rapid dissemination and utilization of existing laws such as those that protect organizations donating food [22]. On the consumer side, the CDC published guidelines on food safety and running essential errands including food shopping. The USDA provided relief to many people struggling to afford food through the Farmers to Families Food Box Program [23]. For this program, the USDA contracted with food distributers and other retail actors to distribute excess farm products (which normally would have gone to restaurants) through emergency food channels. A variety of existing federal nutrition education and promotion materials have been disseminated, modified, or created during the pandemic, particularly around food safety [24,25,26]. The USDA denied waivers from several states to use SNAP Education (SNAP-Ed) funding to pay for staff to perform work for other federal programs such as school meal distribution [27]. SNAP-Ed is an evidence-based program that works to promote healthy eating at the community, state, and tribal levels by using policies, systems, and environmental supports, providing direct nutrition education, and supporting social marketing campaigns [28].

Early in the pandemic, Congress made unprecedented short- and long-term changes to federal nutrition assistance [29]. A key change for the USDA Supplemental Nutrition Assistance Program (SNAP), which provides funding to supplement the food budgets of income-eligible individuals and families, was the expansion of online purchasing, which is now available in 45 states and the District of Columbia, impacting more than 90% of SNAP participants [30,31]. More work remains to expand the SNAP-authorized retailers beyond Walmart and Amazon and to ensure proper protections against predatory exposures to unhealthy food marketing [32,33,34,35]. Despite increasing access to online shopping, SNAP benefits have not yet been increased [32]. However, states could request waivers from the USDA to provide additional SNAP benefits through emergency allotments (up to USD 646 for a family of four) through Pandemic Electronic Benefits Transfer (P-EBT) for households with children who would normally receive free or reduced-price school meals (estimated USD 114 per child per month) [36]. The Special Supplemental Nutrition Program for Women, Infants, and Children (WIC) regularly provides supplemental foods, health care referrals, and nutrition education for income-eligible pregnant, breastfeeding, and non-breastfeeding postpartum women, and to infants and children up to age five who are found to be at nutritional risk. Online food payment is not currently possible with WIC benefits, though workarounds are available to order online with curbside payment and pickup [37,38]. A recent brief detailed key COVID-19 provisions to help optimize program impacts, including modernizing and streamlining WIC enrollment, extending eligibility for mothers and children, enhancing and expanding outreach, examining WIC food package flexibilities, scaling up nationwide best practices, and evaluating changes in breastfeeding practices during the pandemic [39].

Tribes, states, and local governments have played significant roles in shaping the RFE including creating food retail capacity and opening restrictions aimed at reducing the transmission of COVID-19. They have also increased technical assistance and communication regarding enrollment in new and existing federal nutrition assistance programs (e.g., P-EBT and Grab-n-Go meals), funding to support emergency feeding programs, and policies and resource allocations that support home delivery to vulnerable populations (e.g., state agencies covering delivery fees for SNAP online purchases) [40,41,42]. To offer Grab-n-Go meals to children during school closures, the USDA granted schools and other community sites flexibility to serve meals that do not meet the National School Lunch and School Breakfast programs’ nutrition standards [43].

## 3. Retail Sources

COVID-19 has had multiple impacts on access to the places and the means by which people obtain food. Perhaps the most notable effects have come from restaurant closures. As about half of America’s food dollars and a third of the food products produced in this country go to food service (food prepared away from home), including both restaurants and institutional food service (e.g., schools, hospital cafeterias), the closing of many of these venues significantly shifted both where Americans get their food and where food supplies are sent (or not sent). March 2020 spending on food away from home was 51% percent lower than it was in March of 2019 [44]. More cooking and eating at home has meant that other sources, like grocery stores and fresh food delivery services, have seen a surge in demand.

Since dollar stores and larger retailers like Target and Walmart sell groceries, they have been able to stay open when other retail outlets have been forced to close and may have seen increased sales as a result. Even as food sales have shifted from prepared food sources to retail food stores, shopping access has been limited by shorter store hours that allowed staff more time for cleaning or by designated shopping times for vulnerable community members (e.g., seniors, immunocompromised) [45]. Grocery workers protesting poor working conditions have also threatened to limit grocery store access [46].

During the initial onset of the pandemic, online ordering and sales, which allow for no or low-contact purchasing, surged with as many as 78.7% of consumers reporting having shopped online (compared with 39% pre-pandemic) [7]. However, regular online shopping rates have remained modest, with 33% of people reporting shopping online at least once a week compared to 27% in 2019 [47,48]. Prior to COVID-19, online grocery shopping rates were highest in the 30–44 age group with 28.3% reporting shopping for groceries online in 2019. This trend, mainly driven by families with children who desire convenience, has continued during COVID-19 [49]. Younger shoppers in general are more likely to embrace the technology needed to shop online than their older counterparts [50]; only 10% of baby boomers report that they will continue shopping for food online after the pandemic is over compared with 35–40% of younger shoppers [51].

Online shopping may also be disproportionately observed among wealthier, urban consumers as opposed to people who have limited income or live in rural communities, who may lack credit cards or reliable internet [52,53]. Federal nutrition assistance benefits are not accepted online at all (in the case of WIC) or only by select retailers (in the case of SNAP). On the business side, the switch to online sales may have left some small businesses behind. Larger supermarkets with existing online ordering capability have more easily adapted to the online order environment, while smaller stores without dedicated e-commerce platforms have been left scrambling to complete phone orders or create homespun website solutions [54].

For restaurants, many delivery services (e.g., Grubhub, Skip the Dishes) are expensive and can cut into already reduced margins [55]. Consumers may assume that restaurants that do not use these services are closed or they may merely overlook them. Local food purveyors such as farmers and mobile markets that directly sell produce and other foods to consumers have also struggled to adapt to online sales and home delivery as they may have limited staff for deliveries or are unable to take orders from customers with limited internet access [56,57]. Regardless of the food source (restaurant, grocery delivery, mobile market), delivery options may be limited, or totally unavailable to many rural consumers due to the extensive distances needed to travel to provide delivery in those areas, further exacerbating disparities in accessing food in rural communities [47].

Finally, food banks and food pantries have seen a huge surge as consumers who lost their jobs have turned to food aid. Emergency food is also an important source for those not eligible for federal nutrition assistance or those who fear the chilling effect of the Public Charge Rule, which threatens the legal status of immigrants who accept certain forms of government assistance [58,59]. At the same time, food banks and pantries were impacted by disruptions in the food supply as well as decreases in donations and volunteers, who tend to be older and more vulnerable to COVID-19.

## 4. Retail Actors

Important retail actors affected by the pandemic include the owners and managers of food retail and food distribution businesses that control what food is available where. The most notable pandemic challenges for food distributors has been that supply chains and products developed for food service and restaurants need to be adjusted to get food to grocery stores and other retail outlets (i.e., packaging sizes are different for stores vs. restaurants) [3]. For many distributors, switching from food service to food retail sales requires flexible packing lines and transportation channels as well as diverse distribution relationships which most of them do not currently have [3]. Without the ability to quickly readjust distribution channels, the shift in demand has led to temporary decreases in availability of many foods at retail outlets (see additional detail in Customer Retail Experience below). However, some smaller producers and distributors were able to transition to selling food directly to the public. In addition to shifting supply, there has been a redirection in the workforce as grocery stores have had to hire more employees and restaurants in turn have laid off workers. The pandemic has also precipitated an increase in “gig” economy positions including food delivery for popular restaurant and grocery delivery apps like Instacart and Grubhub [60]. Despite being labeled as essential workers, many food retail and delivery jobs have limited benefits such as hazard pay or sick leave and may leave these workers vulnerable to food insecurity themselves [61].

## 5. Business Models

Many retail food businesses have been forced to change and adapt their business models to both serve the needs of their communities and ensure their survival. In addition to the rapid expansion of online shopping and delivery services observed in grocery stores and restaurants, many retail food businesses have transitioned their products or target markets. For example, some restaurants have transitioned to mission-driven work (e.g., providing food for hospital workers or laid-off restaurant workers) and are relying on donations from customers to keep their businesses afloat [42]. Others have pivoted by creating prepare-at-home food or by opening a farm stand or mobile grocery store which has allowed them both to sell-off excess stock and take some of the burden off of over-crowded grocery stores [62]. Many restaurants have enhanced outdoor seating thanks to parking lots and sidewalks that were turned into make-shift patios.

While the large majority of grocery stores remained open with adjustments, some non-traditional retailers (mobile markets, farmers’ markets, community-supported agriculture) initially shut down over fears of not being able to safely serve their customers, especially those who served primarily senior populations [44]. This has resulted in increased sales for grocery stores while 74% of farmers’ markets say that they have lost income [63]. Other farmers and small markets have found ways to stay open by converting to pre-packaged foods (i.e., bundle or box models) that eliminate having customers touch the food or spend extra time making selections. Many small stores (corner stores, bodegas, etc.) in urban settings were already outfitted with plexiglass partitions and had existing practices that limited the number of clients entering the store; therefore, these small stores were paradoxically better prepared for the pandemic in some ways compared to larger food stores [64,65]. There has also been a rise in direct sales by producers including food manufacturers’ bulk shipping of canned goods, snacks, and other shelf-stable items directly to customers [66]. Notably, in May 2020, PepsiCo launched Snacks.com and PantryShop.com to sell its products directly to consumers. On a smaller scale, farmers and fishermen started community-supported farm or fishery programs [62].

## 6. Customer Retail Experience

COVID-19 has had an unprecedented impact on all aspects of the customer retail experience, most notably availability and prices. Customers have reported having to visit multiple retailers to find desired foods and beverages or say they are not able to find the types of foods and beverages their families prefer [67]. These reductions in food availability disproportionately impact communities that already had reduced access to retail food sources like rural communities and communities of color [68]. Supply chain issues limited the availability of many foods and beverages, including infant formula, but empty grocery store shelves were more a result of consumer behavior [69]. Many consumers started stocking up on food, either because they were afraid they would not be able to find the items or wanted to limit the number of times they had to leave their home [70]. These customer behaviors inadvertently deepened inequities in food access. People with limited income do not generally have enough disposable income to make bulk purchases and may be limited in how much they can buy if they rely on public transit. Limited stock may force them to make multiple trips or rely on more expensive small stores closer to home. Although flexibilities were available in some states due to the pandemic, WIC participants are limited to shopping in WIC-approved stores, are generally limited to specific sizes and brands, and cannot substitute when a WIC-approved brand is sold out, essentially making them unable to redeem much-needed benefits.

The shift to more grocery shopping and at-home preparation has led to some price increases (Table 3). In April 2020, the cereal and bakery index saw the largest monthly price increase ever recorded by the Bureau of Labor Statistics (3.1%) [71]. In May 2020, consumer prices for meats and eggs rose 10% which was the largest 12-month percentage increase since 2004; these increases were partially due to changes in meat supply due to COVID outbreak-related closures in meat processing facilities [71]. In addition to higher prices, consumers have seen fewer promotions and advertisements during the pandemic as stores are already dealing with increased demand and want to decrease traffic. Some food companies even pledged not to offer any promotions for at least the initial pandemic months [72]. However, marketing may have shifted to different venues; Kraft Heinz, Kellogg’s, and McDonald’s were forced to temporarily cease advertising on online learning platforms after advocacy groups raised concerns over ads for unhealthy foods being advertised to children [73].

## 7. The Customer and Individual Dietary Intake

The RFE changes described here have likely had profound and lasting effects on the shopping behavior and diets of customers. Customers have reduced in-person shopping frequency; only 20% of customers reported multiple shopping trips each week (down from 28% in 2019) [44]. Preliminary data collected during the pandemic indicates increases in cooking at home, following a diet, snacking, and eating plant-based foods [47]. Importantly, these effects appear to be differentially felt depending on the individual, interpersonal, and household level characteristics of each customer. Individuals or households who live in communities with greater access to a variety of food sources, and who have sufficient resources, steady employment, and credit are more able to adapt to the pandemic food environment. In addition to financial savings, the positive benefits of transitioning to fewer prepared foods may include consumption of more locally produced foods (especially local produce) and decreased reliance on ultraprocessed foods [75,76].

Changes in food intake are likely very different among the economically and nutritionally vulnerable members of our society. Individuals experiencing food insecurity are more likely to buy less expensive and less healthy foods and beverages, such as packaged and ultraprocessed items. A study of the early effects of COVID-19 indicated that 41% of food-insecure individuals reported buying fewer fresh items (i.e., milk, meat, fruits and vegetables) compared to 21% of food-secure individuals [77]. Food-insecure individuals are also more likely to need to access other components of the food system, including food pantries and free meal distribution sites, such as school meal distribution. Lower diet quality associated with food insecurity could potentially exacerbate already higher rates of diet-related diseases that resulted in part from historic structural barriers in food access [15,78]. This is especially problematic as diet-related diseases, such as hypertension, obesity, and diabetes, are associated with COVID-19 hospitalizations and higher mortality [79,80].

## 8. Future Directions

The COVID-19 pandemic has put unexpected strains on our nation’s food system, upending many traditional food supply and access strategies employed by retailers and customers, and establishing new ones. It will not only be important to study the direct impact of changes to the RFE during the pandemic, but to look at the implications for building more resilient food systems following the pandemic. This work in particular should focus on our most under-resourced community members, including low-income communities and communities of color, whose access to healthy foods was already limited. Recommendations for further research at the consumer, retail, community, and policy levels are outlined.

Little is yet known about the effects of the pandemic on consumer behavior, including shopping frequency and the types of foods purchased. Fewer shopping trips due to pandemic exposure concerns may have led to less purchasing of fresh foods, such as fruits and vegetables [81]. It is likely that far greater quantities of ultraprocessed and long-term storage foods were purchased, but the longer-term impacts of these food system shifts on the diet of households and individuals are not known. Future research is needed to understand how changes in shopping patterns due to the pandemic have affected what people buy and eat. The pandemic accelerated an existing trend towards online ordering of both groceries and prepared foods. Research on online food purchasing behaviors is still nascent and there is an urgent need to better understand who is buying online and from what types of sellers (i.e., retailers, manufactures). We also need to better understand the role of marketing and other behavioral economic factors, especially given the reports of increased surveillance and marketing of unhealthy foods to SNAP participants shopping online [32,33]. It is also critical that accessibility needs for underserved populations, including rural customers and SNAP and WIC recipients, be better understood and addressed.

The movement of food retail towards alternative (and in some instances, more community-minded) models combined with the closure of many restaurants underscores a need for research looking at business sustainability, especially in lower-income and minority communities. Food retail already operates on very slim margins and the US has some of the cheapest food in the world in relation to income with low costs coming at the expense of small farmers, food workers, and our environment. Research to understand what it will take to change this exploitive relationship is needed. This is particularly relevant as business interest expands outside profit to include public health, supporting local producers and economies, and simultaneously being able to advance equity through affordable pricing and paying a living wage. Outlining how private companies can meet these goals will help retail food actors move past the purely profit-driven model which has contributed to current health disparities. More diversity within business models and retail actors (e.g., minority and immigrant-owned businesses) who live and support the communities they serve may be one step towards this goal. Research is needed to understand how the effectiveness of existing policies and community programs (e.g., business incubators, healthy food financing, public procurement) can support business diversity at the local level. We also need more community-engaged research to understand the impact of food policy councils and other forms of community representation, particularly among Indigenous peoples and communities of color.

At the federal level, we need to advance our understanding of the impacts on families and retailers, among others, of the quick and large-scale expansion of federal nutrition assistance. While there has been an increase in SNAP enrollment and some SNAP benefits, current benefit levels are inadequate and legislative attention is needed to define, calculate, and provide adequate SNAP benefits, especially since increasing benefits has been shown to help stabilize the economy [32]. Understanding how flexibilities made to programs like WIC and SNAP in certain states affected program effectiveness can help with creating future resilience and understanding which adaptations should remain during “normal times” [13,82]. We also need to understand the impact of the USDA Farmers to Families Food Box program on both food security and the broader RFE. It is still unclear how this large influx of free food affected small retailers and distributors in lower-income communities.

In thinking about how we can build resilience in our food system and be better prepared for future crises, one possible food system adaptation may be referred to as “smaller is better.” This would include enhancing and supporting local production and shorter supply chains [83]. Although more research is required, smaller enterprises may respond to market disruptions more effectively as they have more ability to shift to new markets and products as they gain insights from an engaged customer base (e.g., the Scale paradox) [84]. We see that when food supply chains are developed to only serve one type of food business (e.g., restaurants), it has ripple effects that head back to the source and negatively affect farmers and producers. Efforts to improve local food control and sovereignty have the potential to enhance food system resilience, but the format and impact of these efforts are just beginning to be explored [85].

## 9. Conclusions

The RFE and Customer Interaction Model provided a useful framework for outlining adaptations and research needs related to the US RFE during the COVID-19 pandemic. Using this model helped highlight vulnerabilities in our food system and future research needs. However, we emphasize that many of the challenges that COVID-19 has brought to the forefront are not new, but instead are the result of a deepening of previously existing inequities, notably in communities of color. Using the model, we were able to identify potential strategies that could help build a more equitable RFE, which may not only benefit our country during normal times but could help build resilience against future pandemics or similar crises.

## Figures and Tables

**Table 1 ijerph-17-07397-t001:** Food insecurity rates before and during COVID-19 for select population groups.

Population Group	Food Insecurity Rate	
	Before COVID	During COVID
All US households [17]	11%	23%
Households with children (<18 years) [20,21]	15%	35%
Mothers with children 12 years and under [20]	15%	41%
Non-Hispanic Black households [18]	21%	38%
Factors contributing to food insecurity during COVID-19: Structural inequities regarding race and class Job loss Holding a low-wage job(s) Limited savings/access to credit		

**Table 2 ijerph-17-07397-t002:** Selected US federal government COVID-19 initiatives targeting the retail food environment and customers *.

**Food Distribution and Donations** *Retail Food Establishments* CDC released guiding principles for restaurants and bars to keep in mind to reduce the risk of COVID-19 spread FDA released the Best Practices for Retail Food Stores, Restaurants, and Food Pick-Up/Delivery Services During the COVID-19 Pandemic and also Best Practices for Re-Opening Retail Food Establishments During the COVID-19 Pandemic—Food Safety Checklist, among other fact sheets and guidance documents Stimulus relief packages included support for retail food establishments (i.e., Paycheck Protection Program, which is a small business loan that helps businesses keep their workforce employed during the COVID-19 pandemic) USDA supported Gus Schumacher Nutrition Incentive Program, which supports projects to increase the purchase of fruits and vegetables among low-income consumers participating in SNAP by providing incentives at the point of purchase, allowed for operational flexibilities during the pandemic, including awarding mini-grants to enable operational changes at farmers’ markets and grocery stores to expand affordable access to fruits and vegetables during this time of need *Charitable Food Network* FEMA Emergency Food and Shelter National Board Program FY 2019 appropriations, which helps provide supplemental funding allocations to local jurisdictions across the country to help support local service organizations that provide critical resources to people with economic emergencies, which include our hungry and homeless populations FEMA public assistance grants, which could be utilized to support emergency food distribution during this pandemic Stimulus relief packages provided increased appropriations and allowed for certain operational flexibilities for The Emergency Food Assistance Program (TEFAP), which helps supplement the food needs of income-eligible Americans by providing emergency food assistance at no cost by providing American-grown USDA Foods and administrative funds to states to operate the program USDA announced the Farmers to Families Food Box initiative, which uses congressional authority to purchase and distribute up to USD 4 billion in agricultural products to food banks and other eligible vendors to distribute to individuals and families in need *Home Delivery* USDA announced partnership with PepsiCo and the Baylor Collaborative on Hunger and Poverty to provide boxes with 5 days of healthy, shelf-stable, individually packaged foods Older American Act, which aims to provide comprehensive funding for critical disease prevention and health promotion services, among other supports such as elder nutrition meal provision, was reauthorized in March 2020 and allowed for flexibilities during this pandemic for drive-through, take-out, or home-delivered meals, providing for grocery delivery, etc. *Export Services* US Department of Commerce announced temporary reductions in or eliminations of costs of several of their export services, which provides relief to US businesses and economic development organizations during this pandemic and encourages the promotion of foreign direct investment and the export of “Made in the USA” foods and beverages around the world during this economic depression *State, Tribal, and Local Governments* FEMA public assistance grants, which could be utilized to support emergency food distribution during this pandemic
**Household Food Handling and Eating Out** CDC released Running Essential Errands, including grocery shopping and take-out CDC released Food and Coronavirus Disease 2019 FDA released and compiled a variety of food safety and COVID-19 resources, including FAQs related to COVID-19 in general and specific to the temporary policy on food labeling
**Federal Nutrition Assistance—Local Access and Purchasing** Stimulus relief packages provided increased appropriations to help with anticipated increased enrollments in the WIC, which provides federal grants to states for supplemental foods, health care referrals, and nutrition education for income-eligible pregnant, breastfeeding, and non-breastfeeding postpartum women, and to infants and children up to age five who are found to be at nutritional risk, and SNAP, which provides nutrition benefits to supplement the food budget of income-eligible families so they can purchase healthy foods and beverages USDA used congressional authority to expand the SNAP Online Purchasing Pilot, with 45 states and the District of Columbia currently participating in the pilot program USDA issued guidance regarding congressionally authorized increased flexibilities during the COVID-19 pandemic such as online enrollment for federal nutrition assistance (i.e., WIC and SNAP, among other programs in the suite of 15 federal nutrition assistance programs) USDA used congressional authority to approve state plans for temporary emergency standards of eligibility and levels of benefits for school children (and now during the extension of this program for school year 2020–2021 to children in childcare) during school and childcare closures, known as P-EBT USDA used congressional authority to provide additional foods for families in the Food Distribution Program on Indian Reservations (FDPIR), which provides USDA Foods to eligible households living on Indian reservations and to American Indian households residing in approved areas near reservations and in Oklahoma
**Federal Nutrition Education and Promotion** USDA developed, modified, or created a variety of federal nutrition education and promotion materials during the pandemic, particularly around food safety and eating on a budget

Note: CDC = Centers for Disease Control and Prevention; USDA = United States Department of Agriculture; FEMA = Federal Emergency Management Agency; FDA = Food and Drug Administration; WIC = Special Supplemental Nutrition Assistance Program for Women, Infants and Children; P-EBT = Pandemic Electronic Benefits Transfer. * Additional tribal, state, and local laws, along with retailer policies and practices, impacted the retail food environment during this pandemic and several other national and international responses impacted the broader food system.

**Table 3 ijerph-17-07397-t003:** Twelve-month percent change in the US Consumer Price Index for food at home Jan–May 2020.

Month	All Items	Food at Home	Cereals and Bakery Products	Meats, Poultry, Fish, and Eggs	Fruits and Vegetables	Dairy and Related Products	Nonalcoholic Beverages and Beverage Materials
Jan 2020	2.5	0.7	0.3	1.9	−1.0	2.7	0.6
Feb 2020	2.3	0.8	0.2	1.9	−1.6	3.6	0.4
Mar 2020	1.5	1.1	0.1	2.3	−1.9	3.7	1.4
Apr 2020	0.3	4.1	3.1	6.8	0.4	5.2	5.0
May 2020	0.1	4.8	2.6	10.0	1.5	5.7	4.1

Note: U S Bureau of Labor Statistics data for urban consumers [74].

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
