# Peer review of "Healthy Food Retail during the COVID-19 Pandemic: Challenges and Future Directions"

_ijerph, 2020, doi:10.3390/ijerph17207397_

Round 1

Reviewer 1 Report

The article discussed the impact of COVID-19 on food retail in the United States (US) and discussed the change in consumer food purchasing behaviour, which affects the sales, distribution and preparation. The article was written well and easy to understand. Introduction was written well and compared the food disparities prior and during the COVID-19 among different types of consumers. The article also discussed the federal, state and community
policies affecting the food retail environment. How different types of people, businesses and their behaviours were changed during the COVID-19 period also discussed. The article also proposed some suggestions for improving / controlling the future of retail food environment. However, there are few minor issues to improve the article (See below). Hence, I recommend the article for publication after a minor revision.

Minor Points:
1. Visualizing the data: If data presented in the graph or picture, it would be easy to follow the readers. Please make a graph to visualize the data, where it is needed. e.g. Line 48-60, Line 141 to 152 or Line 226-237. Or combine the above-mentioned data to make
one graph.
2. Line 145-147: “Online shopping may be disproportionately seen among younger, wealthier, urban consumers as opposed to people who are older, have limited income, or live in rural communities, who may lack reliable internet or credit cards [41,42].” –Why may be disproportionally seen among younger? – are these reference articles presented any numbers? If so present these numbers.
3. Line 196-197: “Grocery stores and other retail food outlets showed significant variation
in how they intially reacted to the pandemic”. – how significant? do you have any numbers? And what is the percentage of the change compared to prior COVID-19.
4. Line 205-207: “There has also been a rise in direct sales by producers including food
manufacturers bulk shipping of canned goods and other shelf stable items from food manufacturers directly to customers” – What is the percentage of change in direct sales?

Author Response

Thank you for the helpful comments.  They have been addressed as follows:

  1. Visualizing the data: If data presented in the graph or picture, it would be easy to follow the readers. Please make a graph to visualize the data, where it is needed. e.g. Line 48-60, Line 141 to 152 or Line 226-237. Or combine the above-mentioned data to make
    one graph.

Response: Thank you for the suggestion.  We have added two new tables: Table 1 at Line 66 provides additional data in reference to the data that was previously in lines 48-60 (food insecurity).  Table 3 at Line 442 provides additional data in reference to the data that was originally in lines 226-237 (food prices).  While we could not find enough consistent data to make a table for the data that was originally in lines 141 to 152 (online shopping) we have provided some additional data as requested below in comment #2.

  1. Line 145-147: “Online shopping may be disproportionately seen among younger, wealthier, urban consumers as opposed to people who are older, have limited income, or live in rural communities, who may lack reliable internet or credit cards [41,42].” –Why may be disproportionally seen among younger? – are these reference articles presented any numbers? If so present these numbers.

Response: We have added additional data at line 293 to address this question.  While we are not aware of any reliable data by age group on online shopping rates pre and post-pandemic, but we note this as needed for future research at line 533.  We have added additional data to this section to clarify the differences that were seen pre-pandemic.  Specifically, we indicate that: “Prior to COVID-19 online grocery shopping rates were highest in the 30-44 age group with 28.3% reporting shopping for groceries online in 2019. This trend, mainly driven by families with children who desire convenience has continued during COVID-19 [49]. Younger shoppers in general are more likely to embrace the technology needed to shop online than their older counterparts [50]; only 10% of baby boomers report that they will continue shopping online after the pandemic is over compared with 35-40% of younger shoppers [51].”

  1. Line 196-197: “Grocery stores and other retail food outlets showed significant variation
    in how they intially reacted to the pandemic”. – how significant? do you have any numbers? And what is the percentage of the change compared to prior COVID-19.

Response: We have removed the line referred to (now 400) as it was redundant with the rest of the paragraph and was not clear as written.  While specific numbers are not currently available to quantify variation, the rest of the section attempts to qualitatively describe some of the different responses that have been seen.

  1. Line 205-207: “There has also been a rise in direct sales by producers including food manufacturers bulk shipping of canned goods and other shelf stable items from food manufacturers directly to customers” – What is the percentage of change in direct sales?

Response: We are not aware of any data quantifying this number, but we note is as a need for future research at line 552. In addition, we offer some additional qualitative evidence at line 412 indicating that in May 2020 PepsiCo launched Snacks.com and PantryShop.com to sell its products directly to consumers.

Reviewer 2 Report

The topic of the impact of the COVID-19 pandemic on healthy food consumption is significant. However, the methodology needs to be seriously reframed and rewrote.

1,  I suggest authors can add paragraphs tilted methods to clearly describe the details of your study.

2, The justification of the Retail Food Environment and Customer Interaction Model used in this study is necessary. 

Author Response

  1. I suggest authors can add paragraphs tilted methods to clearly describe the details of your study.

Response: We apologize for the confusion over format as this was developed as a commentary for a special journal issue on Retail Strategies to Support Healthy Eating and is not a research or review paper. As a commentary does not have a specific methodology we did not include a section as such. Although the format does not match with the traditional intro/methods/results discussion of a research paper it has been approved by the editors for this special issue and we believe it is the best format for conveying the message of manuscript.

  1. The justification of the Retail Food Environment and Customer Interaction Model used in this study is necessary. 

Response: We have justified the use of the chosen model in the introduction (Line 74) as it is currently the only model characterizing the interaction between the retail food environment and dietary intake. 

Reviewer 3 Report

This paper is timely, well-written and structured. Study results may add to the existing knowledge. However, the following suggestions may further enhance its readability:

  1. Duplicated (the same as in the paper title) keywords, such as COVID-19, may not be needed.
  2. Acronym names, such as EBT, FAQs, WIC, SNAP, should be defined only once when first appear and be used thereafter.
  3. To better serve the international readers, brief descriptive of the mentioned US Food Programs may be helpful.
  4. Some of the statistical variations (percentages) pre- and after COVID-19 may be tabulated to enhance more profound emphases.
  5. A typing error in Line 250, “likley”.
  6. A spacing error in Line 285, “ofand”.
  7. Some future customer directions may be helpful also.

Author Response

  1. Duplicated (the same as in the paper title) keywords, such as COVID-19, may not be needed.

Response: We did not see any specific instructions related to this so we will leave it to the journal’s editorial staff to determine if they should be removed.

  1. Acronym names, such as EBT, FAQs, WIC, SNAP, should be defined only once when first appear and be used thereafter.

Response: We have removed duplicate definitions from throughout the paper

  1. To better serve the international readers, brief descriptive of the mentioned US Food Programs may be helpful.

Response: We have added several additional descriptors of programs within Table 2.  In addition, we have made significant additions at lines 136-155 and 174-263 to better describe the United States programs and departments involved.

  1. Some of the statistical variations (percentages) pre- and after COVID-19 may be tabulated to enhance more profound emphases.

Response: We have added tables 1 (on food security) and 3 (on food prices) to more clearly show differences for available data.

  1. A typing error in Line 250, “likley”.

Response: Corrected

  1. A spacing error in Line 285, “ofand”.

Response: Corrected

  1. Some future customer directions may be helpful also.

Response: Additional future directions have been added between lines 526-555 in response to comments by this reviewer and others.

Round 2

Reviewer 2 Report

Thank you for inviting me to review this paper. The revised version is sound and valuable to understand consumer behavior.